# National and subnational burden of leukemia and its risk factors, 1990–2019: Results from the Global Burden of Disease study 2019

Amirhossein Poopak[1☉], Sahar Saeedi Moghaddam[1☉], Zahra Esfahani[1,2], Mohammad Keykhaei[1,3], Negar Rezaei[1,4], Nazila Rezaei[1], Mohammad-Mahdi Rashidi[1], Naser Ahmadi[1], Mohsen Abbasi-Kangevari[1], Mohammad-Reza Malekpour[1], Seyyed-Hadi Ghamari[1], Shirin Djalalinia[5], Seyed Mohammad Tavangar[6], Bagher Larijani[4], Farzad Kompani[7] *

1 Non-Communicable Diseases Research Center, Endocrinology and Metabolism Population Sciences Institute, Tehran University of Medical Sciences, Tehran, Iran, 2 Department of Biostatistics, University of Social Welfare and Rehabilitation Sciences, Tehran, Iran, 3 Feinberg Cardiovascular and Renal Research Institute, Northwestern University, School of Medicine, Chicago, IL, United States of America, 4 Endocrinology and Metabolism Research Center, Endocrinology and Metabolism Clinical Sciences Institute, Tehran University of Medical Sciences, Tehran, Iran, 5 Deputy of Research and Technology, Ministry of Health and Medical Education, Tehran, Iran, 6 Department of Pathology, Shariati Hospital, Tehran University of Medical Sciences, Tehran, Iran, 7 Division of Hematology and Oncology, Children's Medical Center, Pediatrics Center of Excellence, Tehran University of Medical Sciences, Tehran, Iran

☉ These authors contributed equally to this work.
* f–Kompani@tums.ac.ir

**Data Availability Statement:** The dataset used during the current study is available on the GBD

## Abstract

### Background

Hematologic malignancies have a great essential role in cancer global burden. Leukemia which two major subtypes based on the onset, is one of the common subtypes of this malignancy.

### Method

For the GBD 2019 study, cancer registry data and vital registration system were used to estimate leukemia mortality. The Meta-Regression-Bayesian Regularized Trimmed (MR-BRT), Cause of Death Ensemble model (CODEm) and Spatiotemporal Gaussian Process Regression (ST-GPR) were used to model our data and estimate each quantity of interest. Mortality to incidence ratios (MIR) were used to generate incidence and survival from mortality rate. Prevalence and survival were used to generate years lived with disability (YLDs). Age-specific mortality and life expectancy at the same age were used to estimate years of life lost (YLLs). The sum of YLLs and YLDs generates DALYs.

### Results

The total national incidence of leukemia increased from 6092 (UI 95%: 3803–8507) in 1990 to 6767 (4646–7890) new cases in 2019. However, leukemia age-standardized incidence ratio(ASIR) decreased from 11.6 (8–14.8) to 8.9 (6.2–10.3) new cases per 100,000 in this

Compare Tool website. (https://vizhub.healthdata.
org/gbd-compare/).

**Funding:** The author(s) received no specific
funding for this work.

**Competing interests:** The authors have declared
that no competing interests exist.

**Abbreviations:** ALL, Acute Lymphoblastic
Leukemia; AML, Acute Myeloid Leukemia; ASDR,
Age-standardized death rate; ASPR, Age-
standardized prevalence rate; ASIR, Age-
standardized incidence rate; BMI, Body-mass
index; CODEm, Cause of Death Ensemble model;
CLL, Chronic Lymphoblastic Leukemia; CLPD,
Chronic lymphoproliferative disorders; CML,
Chronic Myeloid Leukemia; CMPD, Chronic
myeloproliferative disorders; DALYs, Disability-
adjusted life years; GBD, Global Burden of Disease;
GATHER, Guidelines for Accurate and Transparent
Health Estimates Reporting; ICD, International
Classification of Disease; MR-BRT, Meta-
Regression-Bayesian Regularized Trimmed; MIR,
Mortality to incidence ratio; NASBOD, National and
Subnational Burden of Diseases, Injuries, and risk
factors; PAF, Population attributable fraction; ST-
GPR, Spatiotemporal Gaussian Process
Regression; TMREL, Theoretical minimum risk
exposure level; YLDs, Years lived with disability;
YLLs, Years of life lost.

exact period. At the national level, deaths from leukemia increased 1.5-fold between 1990 and 2019, from 3287 (2284–4201) to 4424 (3137–5030), whereas the age-standardized death rate(ASDR) decreased from 8.3 (6.1–9.8) in 1990 to 6 (4.3–6.8) per 100,000 in 2019. In the study period, total leukemia DALYs decreased 12.2% and reached 162850 (110681–188806), in 2019. The age-standardized DALYs decreased 36.7% from 324.3 (224.8–413.4) in 1990 to 205.3 (140.3–237.8) in 2019. ASDR, DALYs, YLLs, and YLDs rate to high BMI was increasing while smoking and occupational exposure to benzene and formalde-hyde were decreasing in the study period.

## Conclusion

This study provided a better understanding of leukemia burden and to reduce controversies of leukemia across Iran. The leukemia status alteration of the country, is trackable.

## Introduction

Iranians had a life expectancy of 79.6 in female individuals and 76.1 in male individuals. The 78.1% of DALYs number were due to non-communicable diseases encountered Iranian health-care system as a new challenge [1]. Cancer and malignancies are accredited for the major portion of these non-communicable diseases.

Hematologic malignancies have a great essential role in cancer global burden. Leukemia is one of the common subtypes of this heterogeneous malignancy. Leukemia is among the ten most common types of cancer worldwide (3.2% of all cancers). Men are more likely to be diag-nosed with leukemia and die from leukemia compared to women. New cases of leukemia did not change significantly worldwide over the past decade, but the mortality rate of leukemia has been experiencing a 1.7% fall every year. It is showed that the lifetime risk of diagnosis with leukemia is 1.6%, and its risk increases by aging [2–4]. Although the mechanism of leukemia is not entirely demonstrated, several factors such as exposure to cancer-causing agents (chemi-cals), smoking, history of radiation therapy or chemotherapy, myelodysplastic syndromes, rare genetic syndromes, family history smoking, and high body-mass index (BMI) are thought to be involved [5–7].

Leukemia has two major subtypes based on the onset: acute and chronic. Acute leukemia has only two subgroups: Acute Lymphoblastic Leukemia (ALL), and Acute Myeloid Leukemia (AML). On the other hand, chronic lymphoproliferative disorders (CLPD) and chronic myelo-proliferative disorders (CMPD) are two subgroups of chronic leukemia which Chronic Lym-phoblastic Leukemia (CLL), and Chronic Myeloid Leukemia (CML) are the most common subtypes of them, respectively [8]. According to these subtypes of leukemia, it has a variety of mortality, incidence, prevalence, and DALYs [9]. The incident cases of leukemia in 2018 was 407,000, worldwide, with 309,000 established deaths. A study has shown between 1990 and 2017, ALL and CML experienced a decrease of incidence while CLL and AML incidences had increased in most countries. In the studied period, higher age-standardized incidence were observed in males. In addition, the incidence of leukemia in people aged above 70 has an inclining trend [10]. The trend of diseases could help us with policy decision making which could result in diseases control.

The Global Burden of Disease (GBD) project illustrates a perspective of global, regional, and national health status by assessing health statistics for several causes over a relatively long-

time period [8, 11]. Here, we report the incidence, prevalence, mortality, disability-adjusted life years (DALYs), years of life lost (YLLs), and years lived with disability (YLDs) associated with leukemia in addition to attributed burden to risk factors at national and subnational levels between 1990 and 2019.

## Materials and methods

In this study, we used the general approach of GBD 2019 to estimate incidence, prevalence, mortality, DALYs, YLLs, and YLDs associated with leukemia [8, 11]. The details of the GBD study and the general process of the burden estimation are further up than the scope of our study. So we review the methods from the GBD 2019 study with a focus on leukemia and the epidemiological quantities of interest. Our study is based on 5 categories of leukemia: Acute Lymphoblastic Leukemia (ALL), Acute Myeloid Leukemia (AML), Chronic Lymphoblastic Leukemia (CLL), Chronic Myeloid Leukemia (CML), and others.

Like GBD 2019, our study complies with the Guidelines for Accurate and Transparent Health Estimates Reporting (GATHER) statement. We used Python version 3.6.2, Stata version 13, and R version 3.5.0 to analyze our data [11]. We estimated incidence, prevalence, mortality, YLDs, YLLs, and DALYs—for five age groups ($<$5, 5 to 14, 15 to 49, 50 to 69, and $\geq$70); males, females, and both sexes combined. This study includes subnational analyses for each epidemiological quantity of interest.

The GBD used the International Classification of Disease (ICD) to map the cause list (including 29 cancer groups) and estimate cause-specific mortality [12]. Mapped ICD-10 codes for new cases (hospital/claim analyses) of Leukemia were: C91-C93.7, C93.9-C95.2, C95.7-C95.92, Z80.6, Z85.6 and Mapped ICD-10 codes for death due to Leukemia were C91-C91.0, C91.2-C91.3, C91.6, C92-C92.6, C93-C93.1, C93.3, C93.8, C94-C95.9 [11, 13]. As GBD 2019 study, MR-BRT (Meta-Regression-Bayesian Regularized Trimmed) was used to model the log of the ratios for each cause by age and sex. Further processed data were modeled using the Cause of Death Ensemble model (CODEm) and Spatiotemporal Gaussian Process Regression (ST-GPR) to estimate each quantity of interest by age and sex. To estimate cause-specific mortality, data were matched to the total all-cause deaths, which is calculated in GBD 2019 population, fertility, and mortality estimates. The cause-specific mortality and data of incidence registration were transformed mortality to incidence ratio (MIR). The incidence estimation was calculated by dividing leukemia-caused death by the MIR. To calculate YLDs, after estimating leukemia prevalence and survival rates, the prevalence was multiplied by specific disability. Deaths were multiplied by the life expectancy at that age to calculate YLLs. The sum of YLDs and YLLs calculated DALYs [14, 15]. DisMod-MR 2.1, a Bayesian meta-regression modeling tool, was used to assure consistency between all measured quantities [11].

The four risk factors for leukemia were smoking, high BMI, and occupational exposure to benzene and formaldehyde were considered as behavioral, metabolic and environmental/occupational. We used published systematic reviews and meta-analyzed these relative risks to estimate relative risk as a function of exposure for each risk-outcome pair. Here in all of our risk factors were dichotomous. For each risk factor, a systematic search was conducted for published studies, household surveys, censuses, administrative data, ground monitor data, or remote sensing data to identify the relative risk of leukemia [16]. All data were fully anonymized before we accessed them. ST-GPR and DisMod-MR 2.1 were used to model the data. Standard deviation (SD) and mean were used to model dispersion and ensemble distribution measures, respectively [17, 18]. The theoretical minimum risk exposure level (TMREL) and 0 were determined as the low point of risk function for J-shaped or U-shaped risk functions and monotonically increasing risk functions. TMREL was generated by distortion of each risk-

outcome pair by outcome relative global magnitude. To compute the population attributable fraction (PAF), we used the formula described in the previously published papers [8]. To prevent overestimating the PAF and the attributable burden for combinations of risks, we used the mediation matrix described in GBD 2017 [19].

Decomposition analysis was employed to find the proportion of population growth, age structure change, and incidence rate change on the overall change of incident cases from 1990 to 2019 [20].We used GBD standard population to calculate the age-standardized rate [21]. Also, each quantity point of estimation is reported with its 95% uncertainty interval (95% UI) to increase the constancy level of this study.

## Ethics

This study was evaluated and approved by Research Ethics Committees of Endocrinology & Metabolism Research Institute, Tehran University of Medical Sciences (Approval ID: IR. TUMS.EMRI.REC.1400.014). All data that were used in this study were fully anonymized before we accessed them.

## Results

### Prevalence

The national prevalent cases of leukemia decreased from 29522 (95% UI: 15810–44076) in 1990 to 28774 (18001–35271) in 2019. Also, leukemia age-standardized prevalence rate (ASPR) decreased 21.0% (-44.4–28.9) in the same period. The prevalent cases of male with leukemia increased 1.3 fold. However, male ASPR as same as female with leukemia showed a decrease (Table 1). In these 30 years, trend of ASPR showed a decrease for first fifteen years and it started to rise after that. ASPR trend showed a constant decrease with a low slope (Fig 1). The majority proportion of ASPR were allocated to leukemia other than AML, ALL, CML, and CLL (Fig 2).

While prevalent cases and prevalence rate of patients below 15 years old showed a significant decrease, prevalent cases and prevalence rate of patients above 15 years old increased either in male or female population. While patients aged below 5 years old had the highest prevalent cases and prevalence rate in 1990, patients aged 15 to 49 became the highest prevalent cases and patients above 70 became the highest prevalence rate in 2019 (Table 2).

Yazd, Fars, and Mazandaran had the highest ASPR among provinces in 2019 whereas Ardebil, Kurdistan, and Khorasan-e-Razavi were the highest in 1990 (Fig 3). The division of highest and lowest ASPR of provinces in 1990 was 2.5 and in 2019 was 2.7, representing an increase in provincial differences (S1 Table).

### Incidence

The national incident cases of leukemia increased from 6092 (3803–8507) in 1990 to 6767 (4646–7890) in 2019. However, leukemia age-standardized incidence rate (ASIR) decreased from 11.6 (8.0–14.8) to 8.9 (6.2–10.3) per 100,000 in this exact period. Men's incident cases of leukemia increased 1.3 fold between 1990 and 2019, though women's incident cases of leukemia decreased 8.6% from 1990 to 2019. The ASIR decreased 33.1% and 15.1% for women and men, respectively (Table 1). The incident cases in the study period showed a decreasing trend till 2002 and a rise after that till 2019. ASIR trend showed a constant decrease in this period (Fig 1). Between 1990 and 2019, incident cases of leukemia increased by 11.1% in Iran which population growth was responsible for 44.0%, age structure change for 0.4%, and an incident

**Table 1. Burden due to leukemia for all ages number and age-standardized rate by sex and year at national level with percent change.**

| Measure | Metric | Year | | | | | | % Change (1990 to 2019) | | |
|---|---|---|---|---|---|---|---|---|---|---|
| | | 1990 | | | 2019 | | | | | |
| | | Both | Female | Male | Both | Female | Male | Both | Female | Male |
| Incidence | All ages number | 6092 (3803 to 8507) | 3197 (1511 to 4674) | 2895 (1429 to 4072) | 6767 (4646 to 7890) | 2923 (1907 to 3374) | 3844 (2208 to 4603) | 11.1 (-21 to 63.1) | -8.6 (-40.2 to 67.1) | 32.8 (-6 to 132.9) |
| | Age-standardized rate (per 100,000) | 11.6 (8 to 14.8) | 11.4 (7.1 to 15) | 11.9 (7.1 to 15.5) | 8.9 (6.2 to 10.3) | 7.6 (5.1 to 8.8) | 10.1 (5.8 to 12) | -23.8 (-40.4 to 5) | -33.1 (-51.7 to -1) | -15.1 (-37.5 to 22.1) |
| Prevalence | All ages number | 29522 (15810 to 44076) | 16950 (6828 to 27307) | 12572 (5260 to 19014) | 28774 (18001 to 35271) | 13136 (8154 to 16061) | 15639 (7873 to 19816) | -2.5 (-36.9 to 74.3) | -22.5 (-54.7 to 65.8) | 24.4 (-23.3 to 154.1) |
| | Age-standardized rate (per 100,000) | 46.7 (28.3 to 65) | 50.9 (25.8 to 74.7) | 42.8 (22.3 to 60) | 36.9 (23.1 to 45.2) | 33.8 (21 to 41.3) | 40.1 (20.1 to 50.6) | -21 (-44.4 to 28.9) | -33.6 (-58 to 20.8) | -6.4 (-40.1 to 55.6) |
| Deaths | All ages number | 3287 (2284 to 4201) | 1509 (893 to 1903) | 1779 (1010 to 2340) | 4424 (3137 to 5030) | 1775 (1195 to 1972) | 2649 (1622 to 3129) | 34.6 (6.6 to 73.7) | 17.7 (-8.3 to 66.4) | 48.9 (14.1 to 129.5) |
| | Age-standardized rate (per 100,000) | 8.3 (6.1 to 9.8) | 7.3 (5.1 to 8.5) | 9.3 (6.2 to 11.5) | 6 (4.3 to 6.8) | 4.8 (3.3 to 5.4) | 7.2 (4.5 to 8.5) | -27.1 (-37.9 to -6.3) | -33.8 (-44.1 to -16.4) | -22.3 (-38.5 to 5.9) |
| DALYs | All ages number | 185562 (120969 to 249702) | 90176 (43279 to 121267) | 95386 (45836 to 132954) | 162850 (110681 to 188806) | 67397 (43013 to 76696) | 95453 (55945 to 114529) | -12.2 (-34.6 to 19) | -25.3 (-46.1 to 32.1) | 0.1 (-28.1 to 70.6) |
| | Age-standardized rate (per 100,000) | 324.3 (224.8 to 413.4) | 308 (181.1 to 389.9) | 339.8 (193.2 to 446.3) | 205.3 (140.3 to 237.8) | 171 (110 to 195) | 239.1 (140.6 to 286.5) | -36.7 (-50.4 to -17.4) | -44.5 (-57.9 to -21) | -29.6 (-47.5 to 4.5) |
| YLLs | All ages number | 182629 (119070 to 246118) | 88624 (42598 to 119594) | 94006 (45230 to 131236) | 159421 (108624 to 184563) | 65891 (42222 to 74935) | 93530 (55058 to 112171) | -12.7 (-34.9 to 18.4) | -25.7 (-46.2 to 31.1) | -0.5 (-28.6 to 69.2) |
| | Age-standardized rate (per 100,000) | 318.8 (221.1 to 407.4) | 302.5 (178.3 to 381.4) | 334.3 (190.2 to 439.5) | 200.8 (137 to 232.6) | 167 (107.9 to 190.1) | 234.1 (137.9 to 280.2) | -37 (-50.7 to -17.9) | -44.8 (-58.1 to -21.4) | -30 (-47.7 to 4.1) |
| YLDs | All ages number | 2932 (1594 to 4567) | 1552 (681 to 2544) | 1380 (602 to 2217) | 3429 (2059 to 4824) | 1506 (849 to 2082) | 1923 (977 to 2787) | 16.9 (-18.6 to 85.9) | -3 (-38.5 to 83.1) | 39.3 (-7.2 to 159.3) |
| | Age-standardized rate (per 100,000) | 5.5 (3.2 to 7.9) | 5.5 (3 to 8) | 5.5 (3 to 8.3) | 4.5 (2.7 to 6.4) | 4 (2.3 to 5.4) | 5.1 (2.6 to 7.3) | -17.7 (-37.4 to 21.8) | -27.8 (-48.9 to 11.7) | -8.3 (-36.4 to 39.3) |

Data in parentheses are 95% Uncertainty Intervals (95% UIs); DALYs = Disability-Adjusted Life Years; YLLs = Years of Life Lost; YLDs = Years Lived with Disability

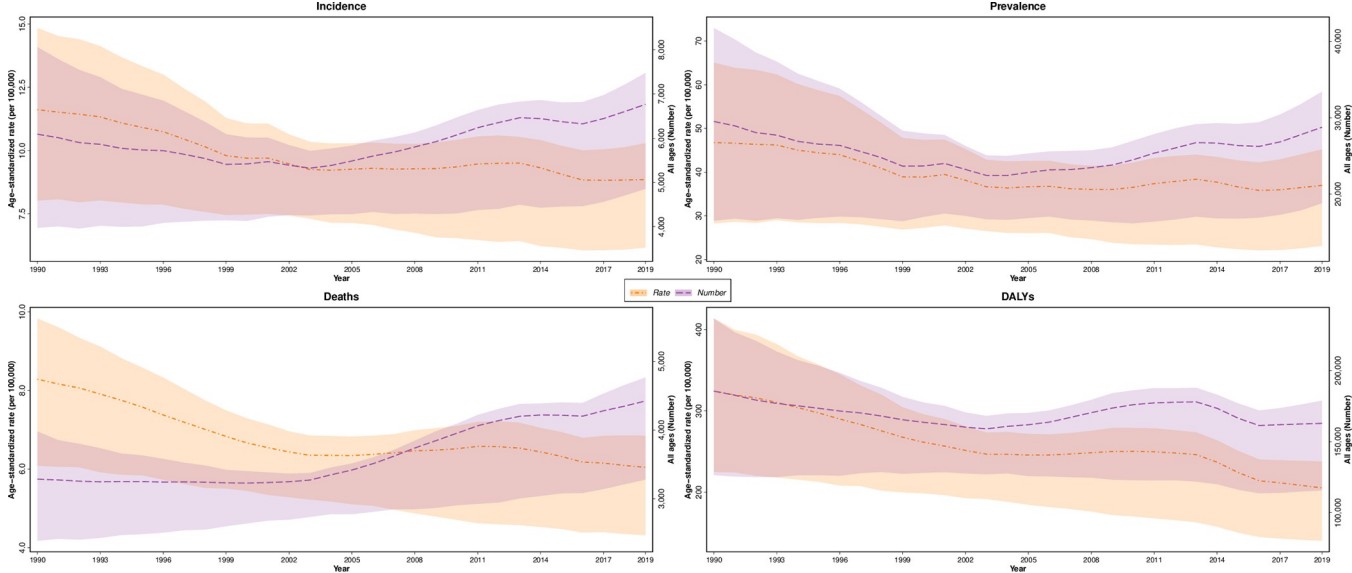

**Fig 1. Time trend of incidence, prevalence, death, and DALYs numbers and age-standardized rates of leukemia, 1990 to 2019 at national level.** Shaded sections indicate 95% uncertainty intervals; DALYs = disability-adjusted life years.

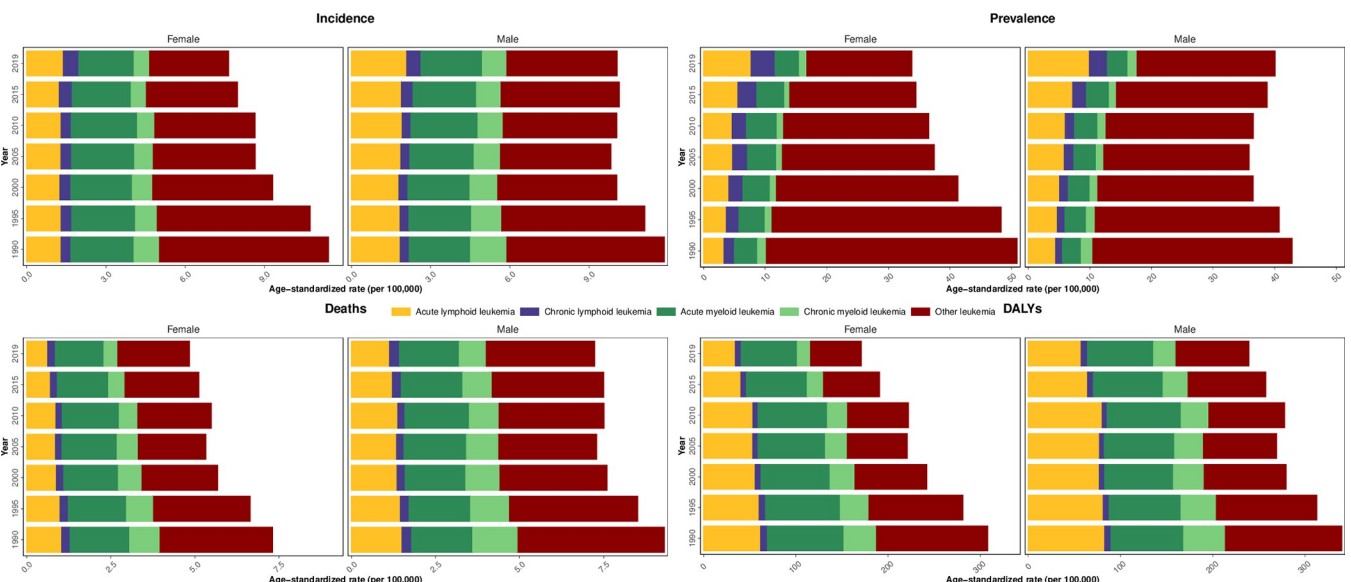

**Fig 2. Proportion of age-standardized incidence, prevalence, deaths, and DALYs rates by leukemia subgroups, 1990 to 2019 at national level.**
DALYs = disability-adjusted life years.

change rate of -33.3%, based on the decomposition analysis (S2 Table). The majority proportion of ASIR were allocated to ALL and AML (Fig 2).

Although people aged 15 to 49 were recorded most incident cases of leukemia in 2019 (2258 [1519–2642]), with a 69.0% (37.0–120.8) increase from 1990, the highest incidence rate in 2019 were occurred among people above 70 years old (39.8 [30.7–44.9]), with 6.3% (-16.7–33.8) increase from 1990 (Table 2).

While Tehran, Khorasan-e-Razavi, and Fars were the three provinces with the highest new cases of leukemia, Fars, Yazd, and Khorasan-e-Razavi showed the highest leukemia ASIR, in 2019 (S2 Table and Fig 3). The division of highest and lowest ASIR of provinces in 1990 was 2.0 and in 2019 was 1.9, representing a slightly decrease in provincial differences (S1 Table).

## Mortality

At the national level, deaths due to leukemia increased 1.3-fold between 1990 and 2019, from 3287 (2284–4201) to 4424 (3137–5030), whereas the age-standardized death rate (ASDR) decreased statistically significant from 8.3 (6.1–9.8) in 1990 to 6.0 (4.3–6.8) per 100,000 in 2019 (Table 1). The death number during this period remained the same till 2002, and it started to increase after that. Though, ASDR showed a decreasing trend in the same period (Fig 1). The highest proportion of ASDR was allocated to AML followed by ALL (Fig 2).

In 2019, leukemia deaths number increased in people aged above 15 and decreased in people aged below 15. Death rate had a decreasing trend in all age groups between 1990 and 2019 except men aged above 70, who had the highest death rate among all age groups, over the study period (Table 2).

Khorasan-e-Razavi remained the highest ASDR between 1990 and 2019 and followed by East Azerbaijan and Fars in 2019 (Fig 3). The division of highest and lowest ASDR of provinces in 1990 was 1.8 and in 2019 was 1.6, hence it shows a decrease in provincial differences (S1 Table).

Leukemia attributed ASDR to smoking decreased statistically significant (-25.8% [-39.2–0.3]), but attributed ASDR to high BMI increased in the study period (25.3% [-1.3–73.8]).

**Table 2. Burden due to leukemia for number and rate by age groups, sex and year at national level with percent change.**

| Measure | Metric | Age (years) | Year | | | | | | % Change (1990 to 2019) | | |
|---|---|---|---|---|---|---|---|---|---|---|---|
| | | | 1990 | | | 2019 | | | | | |
| | | | Both | Female | Male | Both | Female | Male | Both | Female | Male |
| Incidence | Number | < 5 | 2200 (1017 to 3747) | 1449 (397 to 2622) | 752 (162 to 1332) | 366 (164 to 604) | 180 (78 to 317) | 186 (55 to 317) | -83.4 (-92.6 to -57.8) | -87.6 (-95.8 to -49.5) | -75.2 (-92 to -8.4) |
| | | 5–14 | 1268 (779 to 1734) | 600 (287 to 800) | 668 (287 to 1008) | 723 (412 to 978) | 331 (197 to 459) | 391 (165 to 556) | -43 (-63.1 to -6.7) | -44.8 (-65 to -3.1) | -41.4 (-67.6 to 16.9) |
| | | 15–49 | 1336 (880 to 1671) | 645 (330 to 808) | 690 (383 to 901) | 2258 (1519 to 2642) | 1018 (619 to 1208) | 1240 (727 to 1535) | 69 (37 to 120.8) | 57.8 (23.1 to 137.4) | 79.5 (34.4 to 151.6) |
| | | 50–69 | 928 (615 to 1104) | 357 (231 to 426) | 571 (352 to 705) | 2039 (1371 to 2355) | 859 (559 to 999) | 1180 (670 to 1446) | 119.6 (88.1 to 188.3) | 140.4 (96.2 to 212) | 106.6 (65.8 to 184) |
| | | ≥ 70 | 360 (271 to 450) | 146 (109 to 209) | 214 (150 to 260) | 1381 (1065 to 1558) | 535 (419 to 607) | 846 (550 to 993) | 284.1 (201 to 383.3) | 267.4 (148.4 to 385.7) | 295.4 (224.4 to 447.8) |
| | Rate (per 100,000) | < 5 | 24.1 (11.1 to 41) | 32.3 (8.9 to 58.5) | 16.1 (3.5 to 28.5) | 5.2 (2.3 to 8.5) | 5.2 (2.3 to 9.2) | 5.1 (1.5 to 8.7) | -78.5 (-90.5 to -45.4) | -83.8 (-94.5 to -34.2) | -68.2 (-89.8 to 17.9) |
| | | 5–14 | 7.6 (4.6 to 10.3) | 7.3 (3.5 to 9.8) | 7.8 (3.3 to 11.8) | 5.4 (3.1 to 7.3) | 5.1 (3 to 7.1) | 5.7 (2.4 to 8.1) | -28.3 (-53.6 to 17.4) | -30.2 (-55.8 to 22.4) | -26.6 (-59.5 to 46.4) |
| | | 15–49 | 5.1 (3.3 to 6.3) | 5 (2.5 to 6.2) | 5.2 (2.9 to 6.7) | 4.8 (3.2 to 5.6) | 4.4 (2.7 to 5.2) | 5.1 (3 to 6.4) | -5.5 (-23.4 to 23.5) | -11.4 (-30.9 to 33.3) | 0 (-25.2 to 40.2) |
| | | 50–69 | 17.8 (11.8 to 21.1) | 14.7 (9.5 to 17.6) | 20.4 (12.6 to 25.1) | 15.5 (10.4 to 17.9) | 13 (8.5 to 15.1) | 18 (10.2 to 22.1) | -12.7 (-25.2 to 14.6) | -11.6 (-27.8 to 14.7) | -11.7 (-29.1 to 21.4) |
| | | ≥ 70 | 37.4 (28.2 to 46.8) | 30.7 (22.9 to 44.2) | 44 (30.8 to 53.5) | 39.8 (30.7 to 44.9) | 30.7 (24 to 34.8) | 49 (31.8 to 57.5) | 6.3 (-16.7 to 33.8) | -0.1 (-32.5 to 32) | 11.4 (-8.6 to 54.3) |
| Prevalence | Number | < 5 | 14370 (5866 to 25292) | 9838 (2636 to 18529) | 4532 (953 to 8201) | 2385 (1023 to 4091) | 1192 (488 to 2202) | 1192 (288 to 2120) | -83.4 (-93.2 to -51.8) | -87.9 (-96.3 to -49.5) | -73.7 (-93.3 to 0.6) |
| | | 5–14 | 6479 (3507 to 9265) | 3193 (1479 to 4565) | 3286 (1285 to 5361) | 4055 (2123 to 5783) | 1933 (1102 to 2911) | 2122 (739 to 3259) | -37.4 (-62.4 to 19.2) | -39.5 (-65.9 to 16.3) | -35.4 (-68 to 38.7) |
| | | 15–49 | 4733 (3057 to 6030) | 2326 (1269 to 3010) | 2407 (1283 to 3245) | 9755 (6493 to 11769) | 4627 (2680 to 5945) | 5128 (2905 to 6666) | 106.1 (60.7 to 196.5) | 98.9 (46.2 to 215.3) | 113 (49 to 233.4) |
| | | 50–69 | 3010 (1882 to 3709) | 1203 (674 to 1483) | 1807 (1061 to 2319) | 8508 (5248 to 10338) | 3714 (2193 to 4610) | 4795 (2430 to 6256) | 182.7 (132.1 to 310.8) | 208.7 (138.4 to 350.1) | 165.4 (89.9 to 316.6) |
| | | ≥ 70 | 931 (642 to 1204) | 390 (252 to 569) | 541 (352 to 673) | 4072 (2790 to 4744) | 1670 (1132 to 1967) | 2402 (1360 to 2927) | 337.4 (235.7 to 489.2) | 328 (177.3 to 524) | 344.1 (237 to 574.5) |
| | Rate (per 100,000) | < 5 | 157.1 (64.1 to 276.5) | 219.7 (58.9 to 413.7) | 97.1 (20.4 to 175.7) | 33.7 (14.5 to 57.9) | 34.7 (14.2 to 64) | 32.9 (7.9 to 58.4) | -78.5 (-91.1 to -37.6) | -84.2 (-95.2 to -34.3) | -66.2 (-91.3 to 29.4) |
| | | 5–14 | 38.6 (20.9 to 55.3) | 38.9 (18 to 55.6) | 38.4 (15 to 62.6) | 30.4 (15.9 to 43.4) | 29.7 (17 to 44.8) | 31.1 (10.8 to 47.7) | -21.3 (-52.7 to 49.9) | -23.5 (-57 to 46.9) | -19.1 (-59.9 to 73.8) |
| | | 15–49 | 17.9 (11.6 to 22.8) | 17.9 (9.7 to 23.1) | 18 (9.6 to 24.2) | 20.6 (13.7 to 24.9) | 19.9 (11.6 to 25.6) | 21.3 (12.1 to 27.7) | 15.3 (-10.2 to 65.8) | 11.7 (-17.9 to 77.1) | 18.7 (-17 to 85.7) |
| | | 50–69 | 57.6 (36 to 70.9) | 49.6 (27.8 to 61.1) | 64.5 (37.8 to 82.7) | 64.7 (39.9 to 78.6) | 56.3 (33.2 to 69.9) | 73.1 (37.1 to 95.4) | 12.4 (-7.7 to 63.3) | 13.5 (-12.3 to 65.5) | 13.5 (-18.8 to 78.1) |
| | | ≥ 70 | 96.9 (66.8 to 125.4) | 82.3 (53.2 to 120) | 111.1 (72.4 to 138.2) | 117.3 (80.4 to 136.6) | 95.7 (64.9 to 112.8) | 139 (78.7 to 169.4) | 21 (-7.1 to 63) | 16.4 (-24.6 to 69.7) | 25.1 (-5.1 to 89.9) |

(*Continued*)

**Table 2.** (Continued)

| Measure | Metric | Age (years) | Year | | | | | | % Change (1990 to 2019) | | |
|---|---|---|---|---|---|---|---|---|---|---|---|
| | | | 1990 | | | 2019 | | | | | |
| | | | Both | Female | Male | Both | Female | Male | Both | Female | Male |
| Deaths | Number | < 5 | 599 (303 to 1000) | 355 (77 to 584) | 244 (51 to 444) | 92 (46 to 146) | 42 (20 to 68) | 50 (19 to 83) | -84.7 (-92.8 to -63.9) | -88.2 (-95.3 to -44) | -79.6 (-91.7 to -17.2) |
| | | 5–14 | 598 (405 to 786) | 268 (124 to 337) | 330 (152 to 462) | 282 (179 to 357) | 120 (76 to 150) | 162 (85 to 216) | -52.8 (-67 to -30.7) | -55.1 (-67.9 to -26.3) | -50.9 (-69.9 to -9.1) |
| | | 15–49 | 909 (599 to 1136) | 430 (220 to 536) | 479 (265 to 618) | 1264 (853 to 1467) | 531 (319 to 602) | 733 (438 to 902) | 39.1 (12.3 to 79.5) | 23.5 (-1.8 to 80.6) | 53.1 (16.3 to 115.6) |
| | | 50–69 | 785 (523 to 935) | 294 (192 to 350) | 490 (302 to 608) | 1361 (927 to 1557) | 548 (347 to 609) | 813 (474 to 971) | 73.4 (48.6 to 125.9) | 86.1 (53.3 to 134.4) | 65.9 (37.3 to 125.2) |
| | | ≥ 70 | 397 (298 to 496) | 161 (122 to 230) | 235 (164 to 287) | 1425 (1089 to 1620) | 534 (419 to 601) | 891 (579 to 1047) | 259.1 (180.8 to 353.5) | 231.1 (119.8 to 334.3) | 278.3 (208.2 to 428) |
| | Rate (per 100,000) | < 5 | 6.5 (3.3 to 10.9) | 7.9 (1.7 to 13) | 5.2 (1.1 to 9.5) | 1.3 (0.7 to 2.1) | 1.2 (0.6 to 2) | 1.4 (0.5 to 2.3) | -80.2 (-90.7 to -53.3) | -84.6 (-93.8 to -27.1) | -73.8 (-89.3 to 6.5) |
| | | 5–14 | 3.6 (2.4 to 4.7) | 3.3 (1.5 to 4.1) | 3.9 (1.8 to 5.4) | 2.1 (1.3 to 2.7) | 1.8 (1.2 to 2.3) | 2.4 (1.2 to 3.2) | -40.6 (-58.5 to -12.8) | -43.3 (-59.4 to -6.8) | -38.5 (-62.3 to 13.8) |
| | | 15–49 | 3.4 (2.3 to 4.3) | 3.3 (1.7 to 4.1) | 3.6 (2 to 4.6) | 2.7 (1.8 to 3.1) | 2.3 (1.4 to 2.6) | 3 (1.8 to 3.7) | -22.2 (-37.2 to 0.4) | -30.6 (-44.8 to 1.4) | -14.7 (-35.2 to 20.1) |
| | | 50–69 | 15 (10 to 17.9) | 12.1 (7.9 to 14.4) | 17.5 (10.8 to 21.7) | 10.3 (7 to 11.8) | 8.3 (5.3 to 9.2) | 12.4 (7.2 to 14.8) | -31 (-40.9 to -10.2) | -31.6 (-43.6 to -13.8) | -29.1 (-41.3 to -3.7) |
| | | ≥ 70 | 41.3 (31 to 51.6) | 34 (25.7 to 48.5) | 48.4 (33.7 to 58.9) | 41 (31.4 to 46.7) | 30.6 (24 to 34.5) | 51.6 (33.5 to 60.6) | -0.6 (-22.3 to 25.5) | -10 (-40.3 to 18.1) | 6.5 (-13.2 to 48.7) |
| DALYs | Number | < 5 | 53087 (26893 to 88355) | 31528 (6896 to 51913) | 21559 (4510 to 39159) | 8135 (4105 to 12910) | 3727 (1795 to 6087) | 4408 (1694 to 7372) | -84.7 (-92.8 to -63.8) | -88.2 (-95.3 to -44.4) | -79.6 (-91.6 to -17.3) |
| | | 5–14 | 48147 (32492 to 63445) | 21591 (10018 to 27185) | 26556 (12145 to 37235) | 22691 (14294 to 28832) | 9665 (6090 to 12153) | 13026 (6815 to 17476) | -52.9 (-67.3 to -30.3) | -55.2 (-68.2 to -26.1) | -50.9 (-70.1 to -8.5) |
| | | 15–49 | 54550 (35747 to 68439) | 25654 (13082 to 31941) | 28896 (15934 to 37466) | 70974 (47894 to 82637) | 29810 (17872 to 33857) | 41164 (24441 to 50685) | 30.1 (4.6 to 68.3) | 16.2 (-8.1 to 70.9) | 42.5 (7.7 to 99.8) |
| | | 50–69 | 23415 (15565 to 27845) | 8866 (5716 to 10501) | 14549 (8892 to 18060) | 41010 (27857 to 46885) | 16507 (10457 to 18347) | 24503 (14214 to 29316) | 75.1 (50.7 to 127.9) | 86.2 (55 to 136.4) | 68.4 (39.1 to 129.1) |
| | | ≥ 70 | 6363 (4701 to 7941) | 2537 (1895 to 3613) | 3826 (2622 to 4672) | 20040 (14985 to 22750) | 7687 (5841 to 8686) | 12353 (7811 to 14566) | 214.9 (147.2 to 300) | 203 (104.3 to 295.9) | 222.8 (161.6 to 355.7) |
| | Rate (per 100,000) | < 5 | 580.5 (294.1 to 966.1) | 704 (154 to 1159.2) | 461.9 (96.6 to 839) | 115.1 (58.1 to 182.7) | 108.4 (52.2 to 177) | 121.5 (46.7 to 203.2) | -80.2 (-90.7 to -53.2) | -84.6 (-93.8 to -27.6) | -73.7 (-89.2 to 6.3) |
| | | 5–14 | 287.1 (193.8 to 378.4) | 263 (122 to 331.2) | 310.3 (141.9 to 435) | 170.2 (107.2 to 216.3) | 148.8 (93.7 to 187.1) | 190.6 (99.7 to 255.7) | -40.7 (-58.9 to -12.4) | -43.4 (-59.8 to -6.6) | -38.6 (-62.6 to 14.6) |
| | | 15–49 | 206.3 (135.2 to 258.9) | 196.9 (100.4 to 245.1) | 215.5 (118.8 to 279.4) | 150.1 (101.3 to 174.8) | 128.5 (77 to 145.9) | 171 (101.5 to 210.6) | -27.2 (-41.5 to -5.9) | -34.7 (-48.4 to -4) | -20.7 (-40 to 11.3) |
| | | 50–69 | 447.8 (297.7 to 532.5) | 365.4 (235.6 to 432.8) | 519.1 (317.3 to 644.3) | 311.8 (211.8 to 356.4) | 250.2 (158.5 to 278.1) | 373.8 (216.8 to 447.2) | -30.4 (-40.1 to -9.4) | -31.5 (-43 to -13.1) | -28 (-40.5 to -2.1) |
| | | ≥ 70 | 662.3 (489.3 to 826.5) | 535.1 (399.7 to 761.9) | 786.3 (538.8 to 960.2) | 577.2 (431.6 to 655.3) | 440.8 (334.9 to 498) | 714.9 (452 to 843) | -12.9 (-31.6 to 10.7) | -17.6 (-44.5 to 7.6) | -9.1 (-26.3 to 28.3) |

(*Continued*)

**Table 2.** (Continued)

| Measure | Metric | Age (years) | Year | | | | | | % Change (1990 to 2019) | | |
|---|---|---|---|---|---|---|---|---|---|---|---|
| | | | 1990 | | | 2019 | | | | | |
| | | | Both | Female | Male | Both | Female | Male | Both | Female | Male |
| YLLs | Number | < 5 | 51981 (26257 to 86673) | 30815 (6665 to 50674) | 21166 (4440 to 38499) | 7945 (4022 to 12606) | 3637 (1764 to 5896) | 4307 (1662 to 7194) | -84.7 (-92.8 to -64) | -88.2 (-95.3 to -43.9) | -79.6 (-91.7 to -17.3) |
| | | 5–14 | 47488 (32097 to 62474) | 21272 (9809 to 26770) | 26216 (12012 to 36779) | 22300 (14137 to 28230) | 9487 (5981 to 11885) | 12813 (6719 to 17095) | -53 (-67.3 to -30.8) | -55.4 (-68.2 to -26.4) | -51.1 (-70.1 to -9.2) |
| | | 15–49 | 53985 (35428 to 67675) | 25379 (12935 to 31695) | 28606 (15764 to 37058) | 69922 (47250 to 81264) | 29326 (17673 to 33327) | 40596 (24083 to 50045) | 29.5 (4 to 67.6) | 15.6 (-8.6 to 70) | 41.9 (7.5 to 99.4) |
| | | 50–69 | 22985 (15290 to 27293) | 8694 (5628 to 10275) | 14291 (8760 to 17723) | 39945 (27244 to 45642) | 16049 (10195 to 17826) | 23896 (13881 to 28624) | 73.8 (49.5 to 126.1) | 84.6 (53.4 to 134.4) | 67.2 (38 to 126.7) |
| | | ≥ 70 | 6190 (4599 to 7726) | 2463 (1843 to 3504) | 3727 (2568 to 4544) | 19309 (14491 to 21968) | 7392 (5648 to 8307) | 11917 (7555 to 14088) | 212 (145.3 to 295.2) | 200.2 (102.4 to 292.2) | 219.8 (159.3 to 352.7) |
| | Rate (per 100,000) | < 5 | 568.4 (287.1 to 947.7) | 688.1 (148.8 to 1131.5) | 453.5 (95.1 to 824.9) | 112.4 (56.9 to 178.4) | 105.8 (51.3 to 171.4) | 118.7 (45.8 to 198.3) | -80.2 (-90.7 to -53.4) | -84.6 (-93.8 to -27) | -73.8 (-89.3 to 6.4) |
| | | 5–14 | 283.2 (191.4 to 372.6) | 259.2 (119.5 to 326.1) | 306.3 (140.3 to 429.7) | 167.3 (106 to 211.8) | 146 (92.1 to 182.9) | 187.5 (98.3 to 250.1) | -40.9 (-58.9 to -13) | -43.7 (-59.8 to -7.1) | -38.8 (-62.6 to 13.7) |
| | | 15–49 | 204.2 (134 to 256) | 194.8 (99.3 to 243.2) | 213.4 (117.6 to 276.4) | 147.9 (100 to 171.9) | 126.4 (76.2 to 143.6) | 168.6 (100 to 207.9) | -27.6 (-41.8 to -6.3) | -35.1 (-48.7 to -4.5) | -21 (-40.1 to 11.1) |
| | | 50–69 | 439.6 (292.4 to 521.9) | 358.3 (232 to 423.5) | 509.9 (312.5 to 632.3) | 303.7 (207.1 to 347) | 243.2 (154.5 to 270.2) | 364.5 (211.7 to 436.6) | -30.9 (-40.6 to -10.1) | -32.1 (-43.6 to -13.8) | -28.5 (-41 to -3.1) |
| | | ≥ 70 | 644.2 (478.6 to 804.1) | 519.4 (388.7 to 739.1) | 765.9 (527.8 to 933.7) | 556.1 (417.4 to 632.7) | 423.8 (323.8 to 476.3) | 689.7 (437.2 to 815.3) | -13.7 (-32.1 to 9.4) | -18.4 (-45 to 6.6) | -10 (-27 to 27.5) |
| YLDs | Number | < 5 | 1106 (445 to 2007) | 713 (192 to 1428) | 393 (79 to 755) | 191 (75 to 358) | 90 (35 to 183) | 101 (26 to 190) | -82.8 (-92.6 to -54) | -87.4 (-96.1 to -48.9) | -74.4 (-93.1 to -3.3) |
| | | 5–14 | 658 (339 to 1018) | 318 (143 to 482) | 340 (126 to 583) | 390 (188 to 617) | 178 (95 to 295) | 213 (78 to 356) | -40.7 (-63.8 to 8) | -44.2 (-67.4 to 4) | -37.4 (-68.3 to 29.3) |
| | | 15–49 | 564 (341 to 830) | 275 (140 to 401) | 290 (146 to 437) | 1052 (630 to 1482) | 485 (257 to 699) | 567 (302 to 844) | 86.4 (47.6 to 154.2) | 76.5 (34.5 to 174.7) | 95.7 (42.2 to 192.8) |
| | | 50–69 | 429 (242 to 617) | 171 (92 to 243) | 258 (137 to 381) | 1065 (615 to 1497) | 458 (244 to 638) | 607 (299 to 883) | 148.2 (107.1 to 243.9) | 167.4 (109.5 to 275.3) | 135.4 (76.7 to 247.6) |
| | | ≥ 70 | 174 (105 to 250) | 74 (44 to 120) | 99 (59 to 142) | 731 (466 to 979) | 296 (183 to 397) | 435 (242 to 597) | 320.8 (225.4 to 460) | 297.1 (155.8 to 458.5) | 338.5 (240.2 to 538) |
| | Rate (per 100,000) | < 5 | 12.1 (4.9 to 21.9) | 15.9 (4.3 to 31.9) | 8.4 (1.7 to 16.2) | 2.7 (1.1 to 5.1) | 2.6 (1 to 5.3) | 2.8 (0.7 to 5.2) | -77.7 (-90.5 to -40.4) | -83.6 (-94.9 to -33.5) | -67.1 (-91.1 to 24.4) |
| | | 5–14 | 3.9 (2 to 6.1) | 3.9 (1.7 to 5.9) | 4 (1.5 to 6.8) | 2.9 (1.4 to 4.6) | 2.7 (1.5 to 4.5) | 3.1 (1.1 to 5.2) | -25.4 (-54.5 to 35.9) | -29.5 (-58.9 to 31.4) | -21.6 (-60.3 to 62) |
| | | 15–49 | 2.1 (1.3 to 3.1) | 2.1 (1.1 to 3.1) | 2.2 (1.1 to 3.3) | 2.2 (1.3 to 3.1) | 2.1 (1.1 to 3) | 2.4 (1.3 to 3.5) | 4.2 (-17.4 to 42.1) | -0.9 (-24.5 to 54.2) | 9 (-20.8 to 63.1) |
| | | 50–69 | 8.2 (4.6 to 11.8) | 7.1 (3.8 to 10) | 9.2 (4.9 to 13.6) | 8.1 (4.7 to 11.4) | 6.9 (3.7 to 9.7) | 9.3 (4.6 to 13.5) | -1.3 (-17.7 to 36.7) | -1.7 (-23 to 38) | 0.6 (-24.4 to 48.6) |
| | | ≥ 70 | 18.1 (11 to 26) | 15.7 (9.3 to 25.2) | 20.4 (12 to 29.1) | 21.1 (13.4 to 28.2) | 16.9 (10.5 to 22.7) | 25.2 (14 to 34.6) | 16.4 (-9.9 to 55) | 8 (-30.4 to 51.9) | 23.5 (-4.2 to 79.7) |

Data in parentheses are 95% Uncertainty Intervals (95% UIs); DALYs = Disability-Adjusted Life Years; YLLs = Years of Life Lost; YLDs = Years Lived with Disability

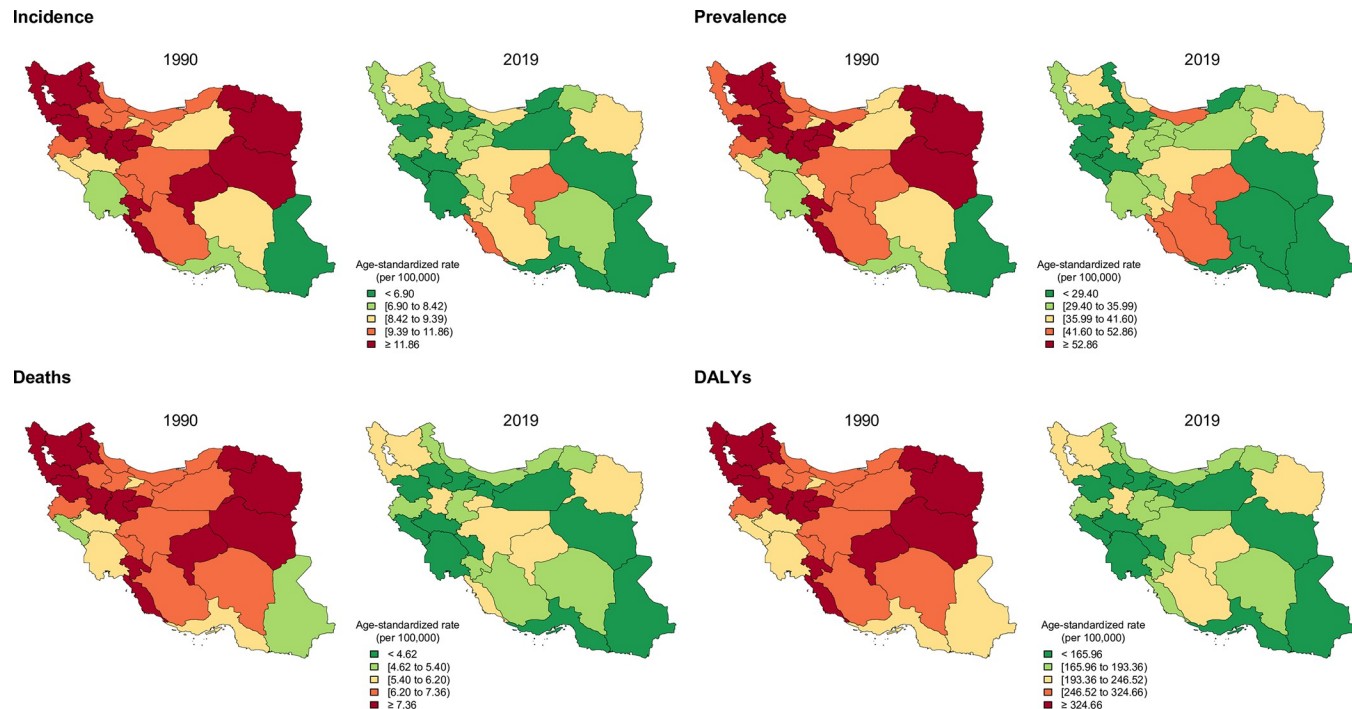

**Fig 3. Geographical distribution of age-standardized incidence, prevalence, deaths, and DALYs rates of leukemia by province, 1990 and 2019.**
DALYs = disability-adjusted life years.

Although leukemia attributed ASDR to occupational exposure to benzene and formaldehyde were inappreciable, they were increased and decreased non-statistically significant in the study period (1.3% [-18.1–33.2] and -16.0% [-34.5–12.5], respectively). The primary risk factor for leukemia attributed ASDR in men and women were smoking and high BMI, respectively (Table 3).

## DALYs, YLLs, and YLDs

In the study period, total leukemia DALYs decreased 12.2% (-34.6–19.0) and reached 162850 (110681–188806) in 2019. The age-standardized DALYs rate decreased statistically significant by 36.7% (-50.4–17.4) from 324.3 (224.8–413.4) per 100,000 in 1990 to 205.3 (140.3–237.8) in 2019 (Table 1). The DALYs trend showed a decrease till 2002 then it increased for nearly ten years, and a decreasing trend started afterward (Fig 1). AML and ALL had the highest allocation of age-standardized DALYs rate (Fig 2).

Leukemia age-standardized DALYs rate were highest in Khorasan-e-Razavi, Yazd, and Hamadan (Fig 3). The division of highest and lowest age-standardized DALYs rate of provinces in 1990 was 2 and in 2019 was 1.7, illustrating a decrease in provincial differences (S1 Table).

Leukemia attributed age-standardized DALYs rate to smoking decreased statistically significant (-30.2% [-42.8–6.0]). However, attributed age-standardized DALYs rate to high BMI increased in the study period (20.3% [-2.8–70.1]). Smoking was a prominent risk factor for leukemia based on the attributed age-standardized DALYs rate in men, and high BMI was a primary risk factor in women (Table 3).

Total leukemia YLLs showed a 12.7% (-34.9–18.4) decrease from 182629 (119070–246118) in 1990 to 159421 (108624–184563) in 2019. Also, the age-standardized YLLs rate decreased

**Table 3. Attributed burden to risk factors of leukemia for all ages number and age-standardized rate by sex and year at national level with percent change.**

| Measure | Metric | Risk factor | Year | | | | | | % Change (1990 to 2019) | | |
|---|---|---|---|---|---|---|---|---|---|---|---|
| | | | 1990 | | | 2019 | | | | | |
| | | | Both | Female | Male | Both | Female | Male | Both | Female | Male |
| Deaths | All ages number | Smoking | 336 (179 to 521) | 37 (13 to 74) | 299 (157 to 466) | 684 (356 to 1021) | 72 (28 to 135) | 612 (314 to 900) | 103.7 (63.9 to 175.7) | 96.6 (36.6 to 205.7) | 104.6 (63.9 to 177.4) |
| | | High body-mass index | 109 (48 to 196) | 69 (25 to 130) | 40 (15 to 79) | 360 (185 to 601) | 197 (79 to 347) | 163 (71 to 289) | 230.6 (166.2 to 363.9) | 185.8 (130.2 to 292.3) | 307.5 (208.7 to 518.3) |
| | | Occupational exposure to benzene | 10 (3 to 18) | 2 (0 to 3) | 9 (2 to 16) | 23 (7 to 41) | 5 (1 to 9) | 18 (5 to 32) | 126.7 (81 to 200) | 205.8 (125.5 to 358.6) | 110.9 (61.8 to 189.7) |
| | | Occupational exposure to formaldehyde | 4 (3 to 6) | 1 (0 to 1) | 4 (2 to 5) | 8 (5 to 11) | 2 (1 to 2) | 7 (4 to 9) | 87.6 (45.8 to 151.9) | 105 (48.7 to 208) | 83.8 (38.1 to 152.9) |
| | Age-standardized rate (per 100,000) | Smoking | 1.3 (0.71 to 2) | 0.29 (0.11 to 0.55) | 2.29 (1.22 to 3.51) | 0.96 (0.5 to 1.44) | 0.2 (0.08 to 0.36) | 1.72 (0.88 to 2.58) | -25.8 (-39.2 to -0.3) | -30.5 (-52.7 to 4.1) | -24.8 (-39.2 to 2) |
| | | High body-mass index | 0.38 (0.17 to 0.69) | 0.49 (0.18 to 0.94) | 0.28 (0.11 to 0.55) | 0.48 (0.25 to 0.8) | 0.52 (0.21 to 0.92) | 0.44 (0.19 to 0.77) | 25.3 (-1.3 to 73.8) | 7.3 (-16.7 to 46.5) | 53.8 (16 to 134.5) |
| | | Occupational exposure to benzene | 0.02 (0.01 to 0.04) | 0.01 (0 to 0.01) | 0.04 (0.01 to 0.07) | 0.02 (0.01 to 0.04) | 0.01 (0 to 0.02) | 0.04 (0.01 to 0.07) | 1.3 (-18.1 to 33.2) | 39.4 (3.7 to 104.6) | -4.1 (-26 to 30.9) |
| | | Occupational exposure to formaldehyde | 0.01 (0.01 to 0.01) | 0 (0 to 0.01) | 0.02 (0.01 to 0.02) | 0.01 (0.01 to 0.01) | 0 (0 to 0) | 0.01 (0.01 to 0.02) | -16 (-34.5 to 12.5) | -6.4 (-30.7 to 35.2) | -16.3 (-36.9 to 15.3) |
| DALYs | All ages number | Smoking | 9635 (4864 to 15416) | 1056 (318 to 2177) | 8579 (4436 to 13524) | 17824 (8751 to 28034) | 1937 (646 to 3906) | 15887 (7755 to 24364) | 85 (51.4 to 150.3) | 83.4 (30.9 to 175.5) | 85.2 (48.1 to 155.7) |
| | | High body-mass index | 3860 (1676 to 6941) | 2504 (908 to 4882) | 1356 (507 to 2718) | 11306 (5642 to 19053) | 6245 (2506 to 11166) | 5061 (2153 to 9050) | 192.9 (133.4 to 324.2) | 149.4 (98.1 to 250.1) | 273.1 (176.5 to 475.7) |
| | | Occupational exposure to benzene | 522 (144 to 921) | 91 (25 to 162) | 431 (118 to 782) | 1135 (324 to 1975) | 259 (75 to 448) | 876 (252 to 1567) | 117.3 (72.5 to 188.1) | 185.1 (107.5 to 337.2) | 103 (54.3 to 181.8) |
| | | Occupational exposure to formaldehyde | 225 (139 to 312) | 42 (22 to 62) | 184 (100 to 264) | 404 (253 to 543) | 80 (47 to 106) | 324 (183 to 457) | 79.1 (38.6 to 142) | 91.2 (37.5 to 194.1) | 76.4 (31.1 to 144.4) |
| | Age-standardized rate (per 100,000) | Smoking | 32.65 (17.49 to 51.32) | 7.45 (2.51 to 15.26) | 56.26 (29.47 to 87.57) | 22.78 (11.62 to 34.54) | 4.96 (1.77 to 9.63) | 40.67 (20.59 to 60.02) | -30.2 (-42.8 to -6) | -33.4 (-53 to -0.7) | -27.7 (-42 to -0.9) |
| | | High body-mass index | 11.17 (4.91 to 20.19) | 14.76 (5.44 to 28.07) | 7.82 (3.01 to 15.44) | 13.44 (6.72 to 22.5) | 14.88 (6 to 26.42) | 12.03 (5.17 to 21.4) | 20.3 (-2.8 to 70.1) | 0.8 (-18.4 to 38.7) | 53.9 (16.3 to 134.7) |
| | | Occupational exposure to benzene | 1.14 (0.31 to 2.02) | 0.39 (0.11 to 0.69) | 1.85 (0.5 to 3.35) | 1.17 (0.33 to 2.04) | 0.54 (0.16 to 0.93) | 1.79 (0.51 to 3.19) | 2.9 (-17.9 to 35.6) | 37.9 (1 to 107.2) | -3.2 (-25.8 to 33.2) |
| | | Occupational exposure to formaldehyde | 0.49 (0.3 to 0.68) | 0.18 (0.09 to 0.26) | 0.79 (0.44 to 1.11) | 0.42 (0.26 to 0.56) | 0.17 (0.1 to 0.22) | 0.66 (0.38 to 0.92) | -15 (-33.7 to 13.9) | -7.3 (-32.4 to 37.8) | -15.8 (-36.4 to 16) |
| YLLs | All ages number | Smoking | 9467 (4774 to 15152) | 1036 (312 to 2135) | 8432 (4350 to 13321) | 17376 (8570 to 27306) | 1883 (629 to 3803) | 15493 (7535 to 23873) | 83.5 (50 to 146.7) | 81.8 (30.1 to 173.8) | 83.8 (46.9 to 153.4) |
| | | High body-mass index | 3801 (1654 to 6836) | 2465 (892 to 4802) | 1336 (500 to 2682) | 11047 (5494 to 18593) | 6094 (2454 to 10868) | 4953 (2106 to 8834) | 190.6 (131.8 to 321.1) | 147.2 (96.1 to 246.5) | 270.7 (174.8 to 472.7) |
| | | Occupational exposure to benzene | 516 (143 to 913) | 90 (24 to 161) | 427 (117 to 775) | 1117 (319 to 1948) | 254 (73 to 441) | 862 (247 to 1535) | 116.2 (71.8 to 186.8) | 183 (106.4 to 333.1) | 102.1 (53.6 to 180.4) |
| | | Occupational exposure to formaldehyde | 223 (138 to 308) | 41 (21 to 61) | 182 (99 to 262) | 397 (248 to 537) | 78 (46 to 104) | 319 (181 to 450) | 78.2 (37.9 to 140.7) | 89.9 (36.5 to 191.9) | 75.5 (30.5 to 143.1) |
| | Age-standardized rate (per 100,000) | Smoking | 32.03 (17.17 to 50.34) | 7.3 (2.46 to 14.9) | 55.18 (28.95 to 85.82) | 22.17 (11.27 to 33.66) | 4.82 (1.72 to 9.34) | 39.61 (20.05 to 58.79) | -30.8 (-43.2 to -6.9) | -33.9 (-53.5 to -1.5) | -28.2 (-42.4 to -1.7) |
| | | High body-mass index | 10.98 (4.8 to 19.8) | 14.5 (5.32 to 27.62) | 7.68 (2.95 to 15.15) | 13.11 (6.59 to 22.08) | 14.5 (5.86 to 25.59) | 11.75 (5.06 to 20.94) | 19.5 (-3.6 to 69) | 0 (-19.1 to 37.8) | 53 (15.7 to 133.3) |
| | | Occupational exposure to benzene | 1.13 (0.31 to 1.99) | 0.39 (0.11 to 0.69) | 1.83 (0.5 to 3.32) | 1.15 (0.33 to 2.01) | 0.53 (0.15 to 0.92) | 1.76 (0.5 to 3.14) | 2.4 (-18.2 to 34.9) | 37 (0.5 to 105.5) | -3.7 (-26.2 to 32.4) |
| | | Occupational exposure to formaldehyde | 0.48 (0.3 to 0.67) | 0.18 (0.09 to 0.26) | 0.78 (0.43 to 1.1) | 0.41 (0.26 to 0.55) | 0.16 (0.1 to 0.22) | 0.65 (0.38 to 0.91) | -15.4 (-34.1 to 13.4) | -7.9 (-32.9 to 36.8) | -16.2 (-36.7 to 15.4) |

*(Continued)*

**Table 3.** (Continued)

| Measure | Metric | Risk factor | Year | | | | | | % Change (1990 to 2019) | | |
|---|---|---|---|---|---|---|---|---|---|---|---|
| | | | 1990 | | | 2019 | | | | | |
| | | | Both | Female | Male | Both | Female | Male | Both | Female | Male |
| YLDs | All ages number | Smoking | 168 (81 to 282) | 20 (6 to 41) | 147 (70 to 246) | 448 (205 to 719) | 54 (18 to 104) | 394 (173 to 644) | 167.2 (111.2 to 281.8) | 167.7 (80.2 to 310.3) | 167.1 (102.8 to 288.4) |
| | | High body-mass index | 59 (24 to 112) | 39 (13 to 79) | 20 (7 to 42) | 259 (124 to 462) | 151 (57 to 284) | 109 (42 to 206) | 338.9 (248.4 to 544.2) | 289 (201 to 467.6) | 433.9 (279.5 to 772.8) |
| | | Occupational exposure to benzene | 6 (2 to 11) | 1 (0 to 2) | 5 (1 to 9) | 19 (5 to 35) | 5 (1 to 9) | 14 (4 to 26) | 218.3 (148.6 to 349.9) | 356.6 (220.9 to 615.4) | 187.8 (108.3 to 326.1) |
| | | Occupational exposure to formaldehyde | 3 (1 to 4) | 0 (0 to 1) | 2 (1 to 3) | 7 (4 to 10) | 1 (1 to 2) | 5 (3 to 8) | 161.3 (97.4 to 275.6) | 204.6 (113 to 382.2) | 151.1 (77.2 to 276.2) |
| | Age-standardized rate (per 100,000) | Smoking | 0.62 (0.31 to 1.03) | 0.15 (0.05 to 0.3) | 1.08 (0.52 to 1.79) | 0.6 (0.28 to 0.98) | 0.14 (0.05 to 0.27) | 1.07 (0.48 to 1.74) | -3.2 (-22.9 to 37.6) | -5.4 (-37.2 to 42.8) | -1.6 (-26.4 to 43.3) |
| | | High body-mass index | 0.2 (0.08 to 0.37) | 0.26 (0.09 to 0.53) | 0.14 (0.05 to 0.28) | 0.33 (0.16 to 0.58) | 0.38 (0.15 to 0.72) | 0.28 (0.11 to 0.53) | 67.7 (32.7 to 142) | 45.9 (13.6 to 110.9) | 104.1 (48.2 to 230.6) |
| | | Occupational exposure to benzene | 0.01 (0 to 0.02) | 0 (0 to 0.01) | 0.02 (0.01 to 0.04) | 0.02 (0.01 to 0.04) | 0.01 (0 to 0.02) | 0.03 (0.01 to 0.06) | 46.3 (15.2 to 106.5) | 111.9 (50.6 to 222.1) | 34.7 (-1 to 99.1) |
| | | Occupational exposure to formaldehyde | 0.01 (0 to 0.01) | 0 (0 to 0) | 0.01 (0 to 0.01) | 0.01 (0 to 0.01) | 0 (0 to 0) | 0.01 (0.01 to 0.02) | 20.4 (-7.6 to 72.3) | 41.7 (0.2 to 118.9) | 17.7 (-15 to 73.1) |

Data in parentheses are 95% Uncertainty Intervals (95% UIs); DALYs = Disability-Adjusted Life Years; YLLs = Years of Life Lost; YLDs = Years Lived with Disability

37.0% (-50.7–17.9) from 318.8 (221.1–407.4) per 100,000 to 200.8 (137–232.6) in the study period (Table 1). The three highest age-standardized YLLs rates were Khorasan-e-Razavi, Fars, and Hamadan in 2019, which Khorasan-e-Razavi stayed province with the highest age-standardized YLLs since 1990 (Fig 3). In the study period, while leukemia attributed age-standardized YLLs rate to high BMI increased (19.5% [-3.6–69.0]), leukemia attributed age-standardized YLLs rate to smoking, and occupational exposure to formaldehyde decreased by -30.8% (-43.2–6.9) and -15.4% (-34.1–13.4), respectively. Smoking with attributed age-standardized YLLs rate of nearly 40 per 100,000 and high BMI with nearly 15 per 100,000 were prominent risk factors in men and women, respectively (Table 3).

While the total YLDs increased 16.9% (-18.6–58.9) in the study period, the age-standardized YLDs rate decreased 17.7% (-37.4–21.8) (Table 1). Yazd, Fars, and Bushehr were the three provinces with the highest age-standardized YLDs rate in 2019 (Fig 3). In the study period, while leukemia attributed age-standardized YLDs rate to high BMI (67.7% [32.7–142.0]) and occupational exposure to benzene (46.3% [15.2–106.5]) and formaldehyde (20.4% [-7.6–72.3]) increased, leukemia attributed age-standardized YLDs rate to smoking decreased by 3.2% (-22.9–37.6). Smoking with attributed age-standardized YLDs rate of 1.07 per 100,000 was a prominent risk factor in men, and high BMI with 0.38 per 100,000 was a prominent risk factor in women (Table 3).

## Discussion

This study used the GBD 2019 study results to analyze the national and subnational burden of leukemia in 31 provinces [8, 11]. Before our study, the mortality of leukemia was assessed in the National and Subnational Burden of Diseases, Injuries, and risk factors (NASBOD) [22]. Their results were different from our study in some aspects because of different methods and data sources. GBD 2019 used mathematical models and the updated cancer mortality to made

estimation on leukemia more robust. The code models and covariates used in those were adjusted for GBD 2019. These modulations make our study more comprehensive and precise.

Our results showed that while the incidence of leukemia is experiencing an increasing trend in Iran, the ASIR is constantly decreasing over the study period. In global studies it has been demonstrated that the incidence of leukemia and ASIR had been observed the same trend as Iran [10]. In comparison with all hematologic malignancies which increased ASIR during 1990 to 2017, our study showed a decrease of ASIR [23]. The total death number of leukemia showed an increase, resulting from population growth, though the ASDR decreased in all provinces. The new chemotherapy regimens over past decades improved leukemia survival in most patients, which could be a probable cause for decreasing trend of ASDR [24–26]. Age-standardized DALYs rate showed a similar declining trend to ASDR since mortality of leukemia generates a significant proportion of leukemia DALYs. Also, the age-standardized YLLs rate decreased, whereas the age-standardized YLDs rate increased between 1990 and 2019. Global results of hematological malignancies supported this study and showed decrease of ASDR and DALYs in general between 1990 to 2017. Attributed age-standardized deaths, DALYs, YLLs, and YLDs rate to high BMI was increasing while smoking and occupational exposure to benzene and formaldehyde were decreasing in the study period. The Global Burden of leukemia has been assessed in a very recent study. They resulted decreased age-standardized rate of DALYs and deaths and increased age-standardized incidence rate, which Iran national and subnational results support global findings except the overall age-standardized incidence rate of hematological malignancies. In which besides leukemia, other hematological malignancies burden were assessed [23].

A significant decrease was observed in age-standardized leukemia incidence, which may have been at least partly driven by reduced exposure to risk factors, high-risk behavioral avoidance, and increased intake of folate and vitamins. This decrease of ASIR in leukemia has been in contrast in all hematological malignancies however, leukemia were exception in the recent global, regional, and national study of hematologic malignancies [23]. The study results were supported by that also, both studies claimed the increase of the incidents which may be due to the fact that new screening strategies has been developed [27, 28]. However, the lesser screening in Iran in comparison with developed countries brought the idea of higher ASIR in Iran which should be further investigated.

Leukemia mortality burden leveled off in this study period, probably after improvements of diagnosis at earlier stages and implementations of newer chemotherapy and treatment regimens [24]. However, in some provinces, non-guideline care is still common. Hence, leukemia outcomes could be improved by adherence to guideline recommendations. It has been shown in previous publications that improving access to healthcare services could control and decrease mortality of leukemia [29]. To further mortality of leukemia decrease, improvements in healthcare infrastructure, accessibility and education are required. Health inequalities between male and female, also between elderly and others should be minimized to improve the outcome of the leukemia and reduce mortality rate even more.

The national and subnational DALYs of leukemia decreased between 1990 and 2019 since the significant proportion of leukemia DALYs is associated with the mortality and YLLs of leukemia. Global study of hematological malignancies supported our findings of DALYs [23]. To reach further success in leukemia DALYs reduction and the burden control, we should first provide health equalities for all people around Iran, in all 31 provinces, with different socio-demographic statuses. Then, earlier diagnosis of leukemia should be prioritized through improving screening practices.

Based on the GBD database, we analyzed smoking, high BMI, and exposure to benzene and formaldehyde as possible risk factors for leukemia. Based on our findings about attributable

risk factors, smoking was the first risk factor for men and the second for women. There are 1.3 billion smokers in the world, of whom men were five times more than women. The higher risk of childhood leukemia with paternal smoking status is reported in a study in 2019 [30]. High BMI was the first risk factor for women. In a cohort study, high BMI was associated with a higher mortality rate [31]. Hyperinsulinemia, high interstitial levels of adipocyte-released fuels, and chronic inflammation due to obesity are probable reasons for poorer outcomes of high BMI [32]. Benzene and formaldehyde, chemical reagents, have different genotoxic effects and chromosomal effects in carcinogenesis [22]. Maternal exposure during pregnancy is demonstrated to increase the risk of leukemia. Also, a higher concentration of benzene and formaldehyde existed in leukemia patients compared to controls [33]. This finding underlined the priority of environmental protection to reduce leukemia outcomes. So the dietary structure and behavioral risk factors should be controlled in order to control leukemia outcomes. Our results showed that in Iran, they were able to reduced exposure to smoking and occupational carcinogens but, poor control BMI in this country is increasing and followed consequences are visible. Hence, further policies and interventions are required to overcome this obstacle.

In addition, the annual repetition of GBD provides the required data and results to find out trends in the burden of diseases over time and hand over data to policymakers to assess the impact of their health programs and policies and compare their results within their region. In each round of GBD study, it improves its estimation by modulating its modeling process and adding new data resources. Innovative policies should take place to reduce the burden of the disease. First of all, no matter of gender, age, and province of residential equality in cancer care for all patients should be provided. All family physicians should be educated the early red flags of leukemia and refer the patients as soon as possible to a hemato-oncologist. Infrastructure of administrating the guideline treatments for every patient should be provided.

## Strengths and limitations

The first and the most important strength of this study was that this study was the first study to address and illustrate the national and subnational burden of leukemia and its risk factors. Furthermore, MR-BRT, CODEm and ST-GPR were used to model our data and estimate the burden of leukemia. This study would help to have a better understanding of leukemia burden and to reduce controversies of leukemia across the country. Several limitations remain obstacles, such as different disease-coding system mapping systems, non-informative or wrongly-coded diseases redistribution, and modeling the inadequate data, which could be addressed by improving methods and registrations.

## Conclusion

In this study, the age-standardized prevalence, incidence, mortality, and disability-adjusted life years (DALYs) rate of leukemia demonstrated a significant decrease over a 30-year period in Iran. The disparities observed between provinces were reduced, with the exception of prevalence. This emphasizes the need for policymakers to prioritize and emphasize access to healthcare and oncological treatments. The reductions observed may be attributed to advancements in treatment approaches and screening programs. It appears that some provinces in Iran have better control and treatment of leukemia, which may be linked to higher income and socioeconomic status, as well as unequal distribution and accessibility of healthcare professionals and systems. The leukemia estimates obtained through the Global Burden of Disease (GBD) study can be used to evaluate progress in cancer management and equity improvements across the country over a 30-year period.

## Supporting information

**S1 Table. Age-standardized rate of leukemia's burden by sex and year at provincial level with percent change.**
(PDF)

**S2 Table. Decomposition analysis of incident cases of leukemia by sex at national and provincial levels.**
(PDF)

## Acknowledgments

We appreciate the aid of all colleagues in Non-communicable Diseases Research Center (NCDRC), and Endocrinology and Metabolism Research Institute, in Tehran University of Medical Sciences

## Author Contributions

**Conceptualization:** Amirhossein Poopak, Sahar Saeedi Moghaddam, Mohammad Keykhaei, Farzad Kompani.

**Data curation:** Sahar Saeedi Moghaddam, Naser Ahmadi.

**Formal analysis:** Amirhossein Poopak.

**Investigation:** Zahra Esfahani, Mohammad-Mahdi Rashidi.

**Methodology:** Sahar Saeedi Moghaddam, Naser Ahmadi.

**Project administration:** Nazila Rezaei.

**Resources:** Mohsen Abbasi-Kangevari.

**Software:** Zahra Esfahani, Mohammad-Reza Malekpour.

**Supervision:** Negar Rezaei.

**Validation:** Mohammad-Mahdi Rashidi, Mohsen Abbasi-Kangevari, Mohammad-Reza Malekpour, Shirin Djalalinia.

**Writing – original draft:** Amirhossein Poopak, Sahar Saeedi Moghaddam.

**Writing – review & editing:** Zahra Esfahani, Mohammad Keykhaei, Negar Rezaei, Nazila Rezaei, Mohammad-Mahdi Rashidi, Naser Ahmadi, Mohsen Abbasi-Kangevari, Mohammad-Reza Malekpour, Seyyed-Hadi Ghamari, Shirin Djalalinia, Seyed Mohammad Tavangar, Bagher Larijani, Farzad Kompani.

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
