## [Decision Letter · Decision Letter 0]

15 Nov 2022

PONE-D-22-23948National and Subnational Burden of Leukemia and Its Risk Factors, 1990 – 2019: Results from the Global Burden of Disease Study 2019PLOS ONE

Dear Dr. Kompani,

Thank you for submitting your manuscript to PLOS ONE. After careful consideration, we feel that it has merit but does not fully meet PLOS ONE’s publication criteria as it currently stands. Therefore, we invite you to submit a revised version of the manuscript that addresses the points raised during the review process.

We look forward to receiving your revised manuscript.

Kind regards,

Mohammad Asghari Jafarabadi

Academic Editor

PLOS ONE

4. We note that Figure 1 in your submission contain map images which may be copyrighted. All PLOS content is published under the Creative Commons Attribution License (CC BY 4.0), which means that the manuscript, images, and Supporting Information files will be freely available online, and any third party is permitted to access, download, copy, distribute, and use these materials in any way, even commercially, with proper attribution. For these reasons, we cannot publish previously copyrighted maps or satellite images created using proprietary data, such as Google software (Google Maps, Street View, and Earth). For more information, see our copyright guidelines: http://journals.plos.org/plosone/s/licenses-and-copyright.

Reviewers' comments:

Reviewer's Responses to Questions

**Comments to the Author**

1. Is the manuscript technically sound, and do the data support the conclusions?

Reviewer #1: Yes

Reviewer #2: No

2. Has the statistical analysis been performed appropriately and rigorously? 

Reviewer #1: No

Reviewer #2: No

3. Have the authors made all data underlying the findings in their manuscript fully available?

Reviewer #1: Yes

Reviewer #2: No

4. Is the manuscript presented in an intelligible fashion and written in standard English?

Reviewer #1: Yes

Reviewer #2: Yes

5. Review Comments to the Author

Reviewer #1: The topic is good and relevant to the Iran. Few things will be needed to be addressed by the authors

Abstract; The conclusion must be improved and linked to the study findings

Background; There is a lack of coherence between the first and subsequent paragraphs. The information presented is inadequate to enable a reader to clearly understand what is the problem/gap and what authors are planning to do. A prior information related to what is known of the burden of leukemia in Iran should be given. This will assist to later establish both the gaps and rationale of the study.

Materials/methods; More details are needed on; data abstraction from the GBD, models and data analysis performed.

Results and discussion; There is a repetition of information, refer line 241-258 and 274-279. No explanation on poor leukemia indicators i.e., high DALYs, YLLs and YLDs among men as compared to women despite a significant reduction in smoking attributed

Some provinces had persistent poor indicators however, no explanation was given behind these.

Conclusion; need to be linked to the results

References; Insert a proper citation for reference number 8

General comment; Insure coherence and improve gramma

Reviewer #2: 1-The trend analysis is a better topic for this investigation. The data that support the conclusions were not included in the manuscript.

2-The statistical models and methods that were implemented to analyze data are not appropriately selected. The multilevel analysis is a better model to analyze such data. It was not mentioned that the Bayesian and spatial models how were implemented, what were priors and posterior functions, how neighborhood structure was selected? How data were generated in the Bayesian method, … ?

3-There was not attached data to the manuscript.

4- The manuscript was presented in an intelligible and technical fashion, however, there is some wrong written phrases ant terms in Standard English, such as; it is showed; in the line 88 in the Introduction section.

5-The tables of the results are very large.

6-The results of the proposed statistical analysis and methods were not presented in tables.

6. PLOS authors have the option to publish the peer review history of their article (what does this mean?). If published, this will include your full peer review and any attached files.

Reviewer #1: No

Reviewer #2: No

---

## [Author Response · Author response to Decision Letter 0]

26 Feb 2023

Response to editor

Dear editor

I hope this letter finds you well. We are grateful about your comments on our manuscript and appreciate your consideration on our manuscript:

1. We ensured that your manuscript meets PLOS ONE's style requirements.

2. The entry data was obtained through the GBD results tool (https://vizhub.healthdata.org/gbd-results/) which is publicly available.

3. Our ethics statement was moved to methods and material section as requested in journal’s requirements.

4. All figures were illustrated using R statistical packages v4.1.0 (http://www.r-project.org/). The map images in figure 1 were created using Open Street Map’s data (without the use of an Open Street Map tile), and the Open Street Map’s shapefile was used in R statistical software to prepare these figures. To respond to your valuable inquiry, we revised the figure caption in the submitted files as “Contains information from OpenStreetMap and OpenStreetMap Foundation, which is made available under the Open Database License”.

5. All the reviewers’ comments were appreciable and helped us to improve the manuscript. We tried our best to brought their thoughtful comments to the manuscript. 

King Regards

---

## [Decision Letter · Decision Letter 1]

22 Mar 2023

PONE-D-22-23948R1National and Subnational Burden of Leukemia and Its Risk Factors, 1990 – 2019: Results from the Global Burden of Disease Study 2019PLOS ONE

Dear Dr. Kompani,

Thank you for submitting your manuscript to PLOS ONE. After careful consideration, we feel that it has merit but does not fully meet PLOS ONE’s publication criteria as it currently stands. Therefore, we invite you to submit a revised version of the manuscript that addresses the points raised during the review process.

 Please submit your revised manuscript by May 06 2023 11:59PM. If you will need more time than this to complete your revisions, please reply to this message or contact the journal office at plosone@plos.org. Please include the following items when submitting your revised manuscript:A rebuttal letter that responds to each point raised by the academic editor and reviewer(s). You should upload this letter as a separate file labeled 'Response to Reviewers'.A marked-up copy of your manuscript that highlights changes made to the original version. You should upload this as a separate file labeled 'Revised Manuscript with Track Changes'.An unmarked version of your revised paper without tracked changes. You should upload this as a separate file labeled 'Manuscript'.If applicable, we recommend that you deposit your laboratory protocols in protocols.io to enhance the reproducibility of your results. Protocols.io assigns your protocol its own identifier (DOI) so that it can be cited independently in the future. For instructions see: https://journals.plos.org/plosone/s/submission-guidelines#loc-laboratory-protocols. Additionally, PLOS ONE offers an option for publishing peer-reviewed Lab Protocol articles, which describe protocols hosted on protocols.io. Read more information on sharing protocols at https://plos.org/protocols?utm_medium=editorial-email&utm_source=authorletters&utm_campaign=protocols.

We look forward to receiving your revised manuscript.

Kind regards,

Mohammad Asghari Jafarabadi

Academic Editor

PLOS ONE

Journal Requirements:

Reviewers' comments:

Reviewer's Responses to Questions

**Comments to the Author**

1. If the authors have adequately addressed your comments raised in a previous round of review and you feel that this manuscript is now acceptable for publication, you may indicate that here to bypass the “Comments to the Author” section, enter your conflict of interest statement in the “Confidential to Editor” section, and submit your "Accept" recommendation.

Reviewer #1: (No Response)

Reviewer #2: (No Response)

2. Is the manuscript technically sound, and do the data support the conclusions?

Reviewer #1: Yes

Reviewer #2: Yes

3. Has the statistical analysis been performed appropriately and rigorously? 

Reviewer #1: Yes

Reviewer #2: Yes

4. Have the authors made all data underlying the findings in their manuscript fully available?

Reviewer #1: Yes

Reviewer #2: Yes

5. Is the manuscript presented in an intelligible fashion and written in standard English?

Reviewer #1: Yes

Reviewer #2: Yes

6. Review Comments to the Author

Reviewer #1: Most of the initial comments have not been addressed.

It is not yet clear on what is known about the burden of leukemia in Iran and why it was important to have their study conducted.

Data abstraction process is still not very clear. A paragraph in the method section talked about the use of systematic reviews but, It is not further shown how authors based their results/discussion on this.

The first paragraph of the strength and limitation section appears more of recommendation.

Still the conclusion is not purely in line with the results.

Reviewer #2: The comments to the authors raised in the previous round of review were not addressed and sent to me. However, there are some grammatical and editing errors in the text, which the authors should correct, as follows:

1- Page 7, line 237, the phrase our stud must be the study or this study

2- Page 7, line 242, the phrase our results must be the results

3- The page of line 259 of the phrase we observed a significant decrease must be a significant decrease was observed.

7. PLOS authors have the option to publish the peer review history of their article (what does this mean?). If published, this will include your full peer review and any attached files.

Reviewer #1: No

Reviewer #2: **Yes: **Solaiman Afroughi

---

## [Author Response · Author response to Decision Letter 1]

31 May 2023

Dear reviewers,

We are grateful about your kind comments. And we did our best to make our article better based on your comments. Below you could find our answers:

Reviewer #1: Most of the initial comments have not been addressed.

It is not yet clear on what is known about the burden of leukemia in Iran and why it was important to have their study conducted.

In the previous revision we checked all the initial comments and in our manuscript with tracked changes you could find them all. 

As we described in the manuscript, the leukemia burden in Iran was not assessed before and there were some disparities in different cities and provinces that should be consider to be changed. So firstly, it was necessary to address the burden of this disease in our country then with our applied statistics we will need to take action to overcome these disparities and also try to reduce the costs of treatments.

Data abstraction process is still not very clear. A paragraph in the method section talked about the use of systematic reviews but, It is not further shown how authors based their results/discussion on this.

We used those systematic reviews to address the most common risk factors of leukemia in our country. And also we used published systematic reviews and meta-analyzed these relative risks to estimate relative risk as a function of exposure for each risk-outcome pair. 

The first paragraph of the strength and limitation section appears more of recommendation.

 Thanks for your precise consideration. There was a small mistake which we corrected and improved it.

Still the conclusion is not purely in line with the results.

 Thanks we tried to improve it based on our results.

Reviewer #2: The comments to the authors raised in the previous round of review were not addressed and sent to me. 

In the previous revision we checked all the initial comments and in our manuscript with tracked changes you could find them all. 

However, there are some grammatical and editing errors in the text, which the authors should correct, as follows:

1- Page 7, line 237, the phrase our stud must be the study or this study

2- Page 7, line 242, the phrase our results must be the results

3- The page of line 259 of the phrase we observed a significant decrease must be a significant decrease was observed.

Thanks for addressing these errors, we corrected them all. 

King regards.

---

## [Editor Report · Decision Letter 2]

15 Jun 2023

National and Subnational Burden of Leukemia and Its Risk Factors, 1990 – 2019: Results from the Global Burden of Disease Study 2019

PONE-D-22-23948R2

Dear Dr. Kompani,

We’re pleased to inform you that your manuscript has been judged scientifically suitable for publication and will be formally accepted for publication once it meets all outstanding technical requirements.

Kind regards,

Mohammad Asghari Jafarabadi

Academic Editor

PLOS ONE

---

## [Editor Report · Acceptance letter]

26 Jun 2023

PONE-D-22-23948R2 

National and Subnational Burden of Leukemia and Its Risk Factors, 1990 – 2019: Results from the Global Burden of Disease Study 2019 

Dear Dr. Kompani:

I'm pleased to inform you that your manuscript has been deemed suitable for publication in PLOS ONE. Congratulations! Your manuscript is now with our production department. 

Kind regards, 

on behalf of

Professor Mohammad Asghari Jafarabadi 

Academic Editor

PLOS ONE